# InCoder: A Generative Model for Code Infilling and Synthesis

**Daniel Fried**[*♡†◇]   **Armen Aghajanyan**[*♡]   **Jessy Lin**[♣]
**Sida Wang**[♡]   **Eric Wallace**[♣]   **Freda Shi**[△]   **Ruiqi Zhong**[♣]
**Wen-tau Yih**[♡]   **Luke Zettlemoyer**[♡†]   **Mike Lewis**[♡]

Facebook AI Research[♡]    University of Washington[†]
UC Berkeley[♣]   TTI-Chicago[△]   Carnegie Mellon University[◇]
dfried@cs.cmu.edu, {armenag,mikelewis}@fb.com

## Abstract

Code is seldom written in a single left-to-right pass and is instead repeatedly edited and refined. We introduce INCODER, a unified generative model that can perform program synthesis (via left-to-right generation) as well as editing (via masking and infilling). InCoder is trained to generate code files from a large corpus of permissively licensed code, where regions of code have been randomly masked and moved to the end of each file, allowing code infilling with bidirectional context. Our model is the first large generative code model that is able to infill arbitrary regions of code, which we evaluate in a zero-shot setting on challenging tasks such as type inference, comment generation, and variable re-naming. We find that the ability to condition on bidirectional context substantially improves performance on these tasks, while still performing comparably on standard program synthesis benchmarks in comparison to left-to-right only models pretrained at similar scale. Our models and code are publicly released.[1]

## 1 Introduction

Large language models trained on vast repositories of code have demonstrated remarkable progress in neural program synthesis and related tasks (Chen et al., 2021a; Austin et al., 2021; Xu et al., 2022; Nijkamp et al., 2022; Chowdhery et al., 2022). However, such models generate code left-to-right, which makes them less directly applicable to many ubiquitous code editing tasks, such as fixing bugs, adding comments, or re-naming variables. We introduce INCODER, a unified model for program synthesis and editing. Like prior work, INCODER is trained to maximize the likelihood of a corpus of code. However, we adopt a causal masking objective (Aghajanyan et al., 2022a), allowing INCODER to infill blocks of code conditioned on arbitrary left and right contexts.

More specifically, we learn to infill by randomly replacing spans of code with a sentinel token and moving them to the end of the sequence (Figure 1, top). The model is trained to predict all tokens in the complete sequence in this permuted ordering. During inference, we can edit code by replacing spans with sentinel tokens, prompting the model with the new sequence, and having it generate new tokens to replace the masked spans (Figure 1, bottom). Because the model can also trivially generate without sentinel tokens, the result is a unified approach for both program synthesis (via left-to-right generation) and editing (via infilling).

We evaluate performance on a range of zero-shot code infilling tasks (Sec. 4), both new and from existing work, including challenging use cases such as type prediction, variable re-naming, comment generation, and completing missing lines of code. Zero-shot infilling with bidirectional context substantially outperforms approaches based on left-to-right-only models, and on several tasks obtains performance comparable to state-of-the-art models fine-tuned on the tasks. Ablation experiments (Sec. 5) show that this does not come at the cost of left-to-right generation ability; our causal masking model achieves similar performance to a standard language model on program synthesis benchmarks (Chen et al., 2021a; Austin et al., 2021) despite its more general training objective.

---

[*]Equal contribution
[1]https://sites.google.com/view/incoder-code-models/

**Training**

Original Document

```python
def count_words(filename: str) -> Dict[str, int]:
    """Count the number of occurrences of each word in the file."""
    with open(filename, 'r') as f:
        word_counts = {}
        for line in f:
            for word in line.split():
                if word in word_counts:
                    word_counts[word] += 1
                else:
                    word_counts[word] = 1
    return word_counts
```

Masked Document

```python
def count_words(filename: str) -> Dict[str, int]:
    """Count the number of occurrences of each word in the file."""
    with open(filename, 'r') as f:
        <MASK:0> in word_counts:
                    word_counts[word] += 1
                else:
                    word_counts[word] = 1
    return word_counts
<MASK:0> word_counts = {}
        for line in f:
            for word in line.split():
                if word <EOM>
```

**Zero-shot Inference**

Type Inference

```python
def count_words(filename: str) -> Dict[str, int]:
    """Count the number of occurrences of each word in the file."""
    with open(filename, 'r') as f:
        word_counts = {}
        for line in f:
            for word in line.split():
                if word in word_counts:
                    word_counts[word] += 1
                else:
                    word_counts[word] = 1
    return word_counts
```

Variable Name Prediction

```python
def count_words(filename: str) -> Dict[str, int]:
    """Count the number of occurrences of each word in the file."""
    with open(filename, 'r') as f:
        word_count = {}
        for line in f:
            for word in line.split():
                if word in word_count:
                    word_count[word] += 1
                else:
                    word_count[word] = 1
    return word_count
```

Docstring Generation

```python
def count_words(filename: str) -> Dict[str, int]:
    """
    Counts the number of occurrences of each word in the given file.

    :param filename: The name of the file to count.
    :return: A dictionary mapping words to the number of occurrences.
    """
    with open(filename, 'r') as f:
        word_counts = {}
        for line in f:
            for word in line.split():
                if word in word_counts:
                    word_counts[word] += 1
                else:
                    word_counts[word] = 1
    return word_counts
```

Multi-Region Infilling

```python
from collections import Counter

def word_count(file_name):
    """Count the number of occurrences of each word in the file."""
    words = []
    with open(file_name) as file:
        for line in file:
            words.append(line.strip())
    return Counter(words)
```

Figure 1: At training time (top), our causal masking objective samples one or more spans of code in training documents (in the upper left figure, a single span) and moves these spans to the end of the document, with their original location denoted by special mask sentinel tokens. An autoregressive language model is trained to produce these entire masked documents, allowing it to learn to generate insertion text conditioned on bidirectional context. At inference time (bottom), we can perform a variety of code editing and infilling tasks in a zero-shot fashion by inserting mask tokens at desired locations and allowing the model to generate code to insert there. All examples shown are real outputs from our INCODER-6.7B model, with the regions inserted by the model highlighted in orange.

## 2 INFILLING AND SYNTHESIS VIA CAUSAL MASKING

Neural models for code generation have either utilized a left-to-right (causal) autoregressive language modeling objective (Brown et al., 2020; Chen et al., 2021a) or, as BERT does, a masked language modeling objective (Devlin et al., 2019; Feng et al., 2020). Both approaches have strengths and weaknesses. Causal models only condition on context to the left of the generated tokens, thus preventing infilling, but they can autoregressively generate entire documents. On the other hand, masked language models can condition on both the left and right contexts to infill a masked region, however, their training objective is typically limited to generating only about 15% of a document. In this paper, we adopt the recently proposed *causal masking* objective (Aghajanyan et al., 2022a), which aims to combine the strengths of both causal and masked language models.

### 2.1 TRAINING

At training time, the causal masking procedure samples a number of *spans* of contiguous tokens in each document to mask (Figure 1, top left). We sample the number of spans from a Poisson distribution with a mean of one, truncated to the support $[1, 256]$, so that there are typically a small

number of spans (with a single span around 50% of the time), but the distribution has a long tail (up to 256 spans). Each span's endpoints are sampled uniformly from the length of the document and the set of sampled spans is rejected and resampled if any spans overlap.

Once spans are sampled, each span $k$ is replaced with a special *mask sentinel token*, `<Mask:k>`. The sequence of tokens in the span is then moved to the end of the document (Figure 1, top right), with the mask sentinel token prepended and a special *end-of-mask* token `<EOM>` token appended. In other words, when a mask token appears for the first time in the left-to-right ordering, it marks the location the span was removed from; when it appears for the second time, it marks the start of the moved span text. More formally, assume we have a document `D` with $N$ tokens, and we have sampled one span `Span = D`$_{i:j}$. Let `Left` be the left context `D`$_{0:i}$ and `Right` be the right context `D`$_{j:N}$. Then, we maximize the log probability of the masked document:

$$\log P([\texttt{Left};\ \texttt{<Mask:0>};\ \texttt{Right};\ \texttt{<Mask:0>};\ \texttt{Span};\ \texttt{<EOM>}]) \tag{1}$$

where ; denotes sequence concatenation. If more than one span were sampled, each would be similarly appended at the end of the document in order. As in standard left-to-right generative language modeling, we compute the probability of the sequence auto-regressively and train the model using cross-entropy loss on all tokens except the mask sentinel tokens `<Mask:k>`, so that the model does not generate these tokens during inference.

## 2.2 INFERENCE

During inference, the model can either be used for left-to-right generation in the standard way (by sampling autoregressively from the model, without using any special tokens), or it can insert code at arbitrary locations in an existing document by inserting a `<Mask:k>` tokens at the desired location(s) and continuing generation at the end of the document. Assuming for simplicity of notation that we want to insert text at only a single location, we can generate a span to insert between the location's `Left` and `Right` context sequences by sampling tokens autoregressively from the distribution

$$P(\cdot \mid [\texttt{Left};\ \texttt{<Mask:0>};\ \texttt{Right};\ \texttt{<Mask:0>}]) \tag{2}$$

until either an `<EOM>` token is generated or a task-dependent stopping criterion is achieved.[2] When applied to code, this allows us to perform tasks that benefit from the bidirectional context in a zero-shot fashion, as shown in Figure 1, bottom. For example, we can perform Python docstring generation conditioned on both the left context (function signature) and right context (function implementation). We can also infill multiple dependent regions, e.g., generate imports required by a function that the model is generating. See Section B.2 for details, including multi-region infilling.

## 3 MODELS

Our primary model is INCODER-6.7B, a 6.7B Transformer (Vaswani et al., 2017) language model. We use the same architecture as the dense 6.7B models described in Artetxe et al. (2021); the Fairseq architecture description can be found in Table 6 in the appendix. All experiments use this model unless stated otherwise (we train smaller models for comparison in Section 5).

To train our models, we collect a corpus of (1) public code with permissive, non-copyleft, open-source licenses from GitHub and GitLab and (2) StackOverflow questions, answers, and comments. Our primary focus in this paper is on the Python language, but we also include code files from 28 total languages and StackOverflow content from all available languages. We decontaminate our pre-training corpus by removing all datasets which we use in our evaluation experiments. See Section A.1 for details. Our final pre-training corpus contains a total of 159 GB of code, 52 GB of it in Python, and a total of 57 GB of content from StackOverflow. See Figure 3 for size by language.

## 4 INFILLING EXPERIMENTS

Our primary evaluation is performing zero-shot infilling for a diverse set of tasks: inserting lines of code, predicting function return types, generating docstrings, renaming variables, and inserting missing code tokens. We formulate each task as filling in one or more masked-out regions of code.

---

[2]In practice, we use a variation of this method that inserts an extra sentinel token into the context to counter a bias on infill length caused by the Transformer's fixed context window size. See Section B.2.

To evaluate how INCODER benefits from bidirectional context when generating infills, we compare three different inference methods: the causal masking inference procedure described in Section 2, a standard left-to-right generation approach (*left-to-right single*), and a left-to-right generation and reranking approach (*left-to-right reranking*). Since our model is also able to generate left-to-right, we can compare all three inference methods using the same INCODER-6.7B model and thus avoid any confounding effects due to a change in the model. For all three inference methods, we obtain generations from the model using top-$p$ (nucleus) sampling (Holtzman et al., 2020) with $p = 0.95$ and a temperature tuned for each task and inference method using the task's development data.[3]

**Left-to-right single.** This baseline does not use the context to the right of the masked location at all. It generates a single completion for the location by conditioning on the left context and sampling tokens autoregressively from the model $P(\cdot \mid \texttt{Left})$ until a task-specific stop condition is reached (e.g., for comment generation, when a comment-ending delimiter is produced).

**Left-to-right reranking.** This baseline uses only the left context to propose candidates to infill the blank, but uses both the left and right contexts to choose among these candidates. Concretely, we first generate $K$ possible completions for the blank region, $\texttt{Span}_1 \ldots \texttt{Span}_K$ following the same procedure as left-to-right single, using $K = 10$ unless otherwise specified. We then evaluate each candidate by substituting it into the blank and scoring the completed document. We use either total log probability of the completed document $\log P([\texttt{Left}; \texttt{Span}_k; \texttt{Right}])$ or, following Chen et al. (2021a), log probability averaged across the number of tokens in the completed document. We select between these two scoring methods for each task using performance on the task's development data.

## 4.1 INFILLING LINES OF CODE (HUMANEVAL)

We create an infilling benchmark for complete lines of code from the HumanEval dataset (Chen et al., 2021a). This dataset provides comment descriptions of functions paired with a canonical implementation of each function and several input–output pairs that the function should pass. HumanEval was introduced as a benchmark for the synthesis of entire Python functions; we evaluate our models on this original synthesis setting in Section C.6. We use this dataset because it affords functional testing of completed code (as opposed to relying solely on an evaluation of the code surface form), which is particularly important when infilling longer regions that have more potential ways to be completed correctly. We construct two infilling tasks from the dataset, for single lines and multiple lines:

**Single-line infilling.** In this task, we mask out each non-blank line of code in the canonical function implementation in turn (creating $N$ examples for a function with $N$ non-blank lines). The task is to generate a single-line completion for the blank conditioned on the natural language description of the function and the code lines before and after the blank. We evaluate using (1) pass rate: the rate at which the completed function passes all of the function's input–output pairs (i.e., analogous to the pass@1 metric from Chen et al. (2021a) and (2) exact match: percentage of times that the completed lines exactly match the masked lines in the canonical implementation. Performance is averaged across all examples generated for all programs in the dataset.

**Multi-line infilling.** This task is constructed in the same way as single-line infilling above but allows each masked region to contain multiple lines of code, creating $N \times (N + 1)/2$ examples for a function with $N$ non-blank lines. We again evaluate completions using pass rate and exact match, averaged across all infilling examples.

**Inference details.** To choose when to end the infill produced by our inference methods, we truncate the candidates generated by the left-to-right (L-R) baselines to the actual number of lines in the blanked-out region. For our causal-masked (CM) infilling method, we end the infill when the model generates the `<EOM>` token. For the L-R single and CM infilling methods, we sample using a temperature of 0.2. For the L-R rerank method, we use a temperature of 0.8 to sample $K = 10$ candidates and rescore with the total log probability of the completed function.

---

[3]For all generation experiments, we prefix prompts with meta-data indicating the code generated should be Python; see Section A.3) for meta-data details.

| Method | Pass Rate | Exact Match | Method | Pass Rate | Exact Match |
|--------|-----------|-------------|--------|-----------|-------------|
| L-R single | 48.2 | 38.7 | L-R single | 24.9 | 15.8 |
| L-R reranking | 54.9 | 44.1 | L-R reranking | 28.2 | 17.6 |
| CM infilling | 69.0 | 56.3 | CM infilling | 38.6 | 20.6 |
| PLBART | 41.6 | — | PLBART | 13.1 | — |
| code-cushman-001 | 53.1 | 42.0 | code-cushman-001 | 30.8 | 17.4 |
| code-davinci-001 | 63.0 | 56.0 | code-davinci-001 | 37.8 | 19.8 |

(a) Single-line infilling.                  (b) Multi-line infilling.

Table 1: On our single- and multi-line code infilling benchmarks that we construct from HumanEval, our causal-masked (CM) approach obtains substantial improvements over left-to-right single candidate and left-to-right reranking baselines in both function test pass rate and exact match.

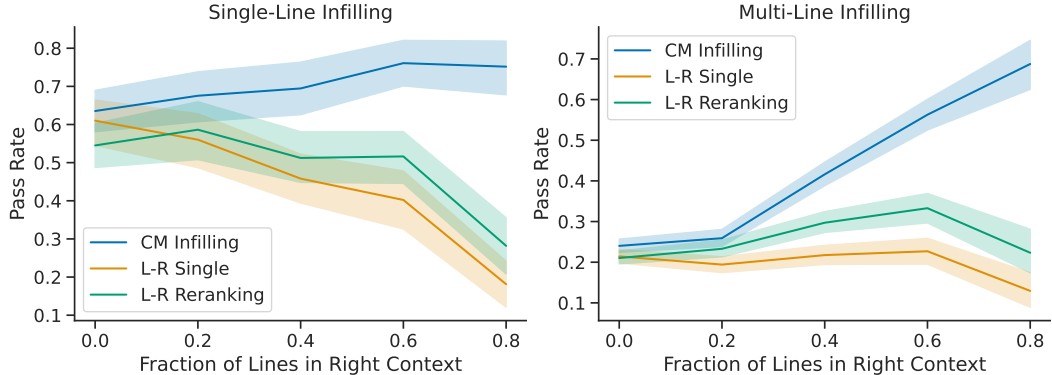

Figure 2: Infilling pass rate by the fraction of the function's lines which are provided to the right of the region that must be infilled, for single-line infilling (left) and multi-line infilling (right). Shaded regions give 95% confidence intervals, estimated using bootstrap resampling. Our causal-masked (CM) infilling method, blue, consistently outperforms both of the left-to-right (L-R) baselines, with larger gains as more right-sided context becomes available (the right side of both graphs).

**Results.** Table 1 shows the results for the single-line (left) and multi-line settings (right). In both settings, CM infilling improves substantially over the L-R single baseline and the L-R reranking baseline. Note that these results are computed by averaging over all examples, which includes masked regions at all positions in functions (including the beginning, when no left context is available, and end, when no right context is available). Figure 2 shows a finer-grained comparison, where we group examples by the fraction of lines in the canonical function which are contained in the context to the right of the infill. The CM infilling method sees larger improvements over the L-R baselines as more right-sided context becomes available (i.e., when the blanked region occurs earlier in the function).[4]

We also compare against two alternate zero-shot methods for incorporating right-sided context: (1) an encoder-decoder code model trained with a denoising infilling objective (PLBART, Ahmad et al. 2021), and (2) templated prompting of large left-to-right generative code models (the cushman-001 and davinci-001 Codex models available through OpenAI's API). See Section C.1 for details on these experiments.[5] InCoder outperforms all models in both single-line and multi-line infilling, despite having lower performance in left-to-right generation than Codex (see Table 11), demonstrating that causal masking training benefits infilling.

---

[4]For multi-line infilling, performance increases with increasing amounts of right context, as having a large right context implies that fewer lines are available to be removed when constructing infill examples, so the resulting examples are often easier to infill.

[5]PLBART outputs tokenized code, which prevents us from computing meaningful exact match metrics.

| Method | BLEU |
|---|---|
| Ours: L-R single | 16.05 |
| Ours: L-R reranking | 17.14 |
| Ours: Causal-masked infilling | 18.27 |
| RoBERTa (Finetuned) | 18.14 |
| CodeBERT (Finetuned) | 19.06 |
| PLBART (Finetuned) | 19.30 |
| CodeT5 (Finetuned) | 20.36 |

Table 2: CodeXGLUE Python Docstring generation BLEU scores. Our model is evaluated in a zero-shot setting, with no fine-tuning for docstring generation, but it approaches the performance of pretrained code models that are fine-tuned on the task's 250K examples (bottom block).

## 4.2 DOCSTRING GENERATION (CODEXGLUE)

We next evaluate documentation string (docstring) generation, where models must generate a natural language docstring that summarizes a Python code snippet. Right context may be particularly useful for docstring generation, as conditioning on the function body can allow models to generate more informative descriptions. Prior neural code generation models are fine-tuned on supervised docstring-code pairs to perform this task (e.g., Clement et al. 2020; Chen et al. 2021a; Lu et al. 2021; Ahmad et al. 2021), however we evaluate our model zero-shot, with no explicit supervision.

We use the CodeXGLUE code-to-text docstring generation task (Lu et al., 2021), which is constructed from CodeSearchNet (Husain et al., 2019), consisting of docstring-code pairs scraped from publicly available GitHub repositories. The L-R single candidate baseline is prompted with the function signature in the left context preceding the docstring. The CM infilling and L-R reranking methods also observe the right context, consisting of the function body.

We compare models following the original automatic evaluation setup for the task. In Table 2, we report smoothed 4-gram BLEU scores for all models, using the reference docstrings provided in the dataset. These references have been preprocessed to strip extraneous content (e.g., argument definitions) from the original scraped docstrings. We use greedy generation for the CM infilling and L-R single candidate generation methods and sample $K = 10$ candidates at temperature 0.8 with average log probability scoring for the L-R reranking method (selected by tuning on the validation set of the task). For all inference methods, we stop generation if the model generates a newline. We also include the performance of the supervised baseline from the CodeXGLUE paper: an encoder-decoder model with a CodeBERT encoder fine-tuned on ∼ 250K training examples from the dataset. Our zero-shot performance approaches the performance of the fine-tuned CodeBERT model.

## 4.3 RETURN TYPE PREDICTION

Predicting return type hints for Python functions is a challenging structured generation task (see Figure 1, "type inference"). We evaluate on two datasets: one we construct from CodeXGLUE and the dataset from TypeWriter OSS (Pradel et al., 2020).

**CodeXGLUE.** We develop a benchmark for return type prediction using the same Python CodeXGLUE dataset used in the code-to-text (docstring generation) task. We run an abstract syntax tree (AST) processor on all functions in the development and test sets of this dataset to (1) identify functions with a PEP 484[6] return type hint annotation that is not None and (2) remove all other type hints (e.g., for function arguments and variable declarations) from the function. This leaves 232 functions in the development and 469 functions in the test set.

The task is to condition on the function signature and body and predict the type hint. We compare the type hints predicted by our various methods to the annotated type hint in the original function, using exact match accuracy on the normalized type hint.[7]

---

[6]https://peps.python.org/pep-0484/

[7]We normalize each type hint by first parsing the type to an AST and then un-parsing the AST to a surface form string, then compute exact match on these surface forms. We note that this metric is somewhat noisy, given

| Method | Accuracy |
|---|---|
| Left-to-right single | 12.0 |
| Left-to-right reranking | 12.4 |
| Causal-masked infilling | **58.1** |

(a) Results on the test set of the benchmark that we construct from CodeXGLUE.

| Method | Precision | Recall | F1 |
|---|---|---|---|
| Ours: Left-to-right single | 30.8 | 30.8 | 30.8 |
| Ours: Left-to-right reranking | 33.3 | 33.3 | 33.3 |
| Ours: Causal-masked infilling | **59.2** | **59.2** | **59.2** |
| TypeWriter (Supervised) | 54.9 | 43.2 | 48.3 |

(b) Results on a subset of the TypeWriter's OSS dataset (Pradel et al., 2020). We include examples from which we were able to obtain source files, successfully extract functions and types, that have non-None return type hints, and that were not included in our model's training data.

Table 3: Results for predicting Python function return type hints on two datasets. We see substantial improvements from causal masked infilling over baseline methods using left-to-right inference.

To compare our three generation methods, we stop generation when a : is generated, which ends the type hint and signals the start of the function body. We tune inference hyperparameters on the development set, and we use a temperature of 0.2 for left-to-right-single, 0.8 for left-to-right reranking, and greedy generation for causal masked infilling. Results on the test set are given in Table 3a. Conditioning on the right context (i.e., the function body) gives some benefit in the left-to-right reranking setting, but gives a substantial improvement via our causal masked infilling.

**TypeWriter OSS.** Some recent work has developed supervised machine learning approaches for predicting type annotations for dynamically-typed languages including Python (Xu et al., 2016; Allamanis et al., 2020; Pradel et al., 2020) and TypeScript (Hellendoorn et al., 2018; Wei et al., 2020; Jesse et al., 2021). We compare our zero-shot model to one such approach for Python, TypeWriter (Pradel et al., 2020), which combines a neural architecture for type hint prediction with a search-based incremental type validation procedure.

To compare to the supervised TypeWriter approach, we obtain its predictions on the open-source software (OSS) dataset used in that work (Pradel et al., 2020), consisting of Python functions from GitHub. Unfortunately, we could not evaluate on their full evaluation set since much of it was included in our model's training data. We filter to instances that were not included in our training data, for which we were able to obtain files and extract functions and types from via AST parsing, and which have non-NONE return type hints. This leaves 2,092 examples (about 12% of their evaluation set). We otherwise emulate their exact setup, which allows our model to condition on file imports, the function body, and the function signature to predict return type hints. We use the same inference hyperparameters as we did for CodeXGLUE type hint prediction.

We present our results in two tables: Table 3b containing metrics across non-None types, and Table 10 in the Appendix, which includes None types as well (following Pradel et al. 2020).[8] We again see benefits from causal masked infilling's ability to condition on the function body when generating return types, and find that our zero-shot model outperforms the supervised TypeWriter model.

## 4.4 VARIABLE NAME PREDICTION

Variable name prediction is a less-constrained code generation task that requires modeling bidirectional context. We again use the test set from the CodexGlue code-to-text task (docstring generation) and run an AST transform to isolate and either mask all the occurrences of the variable name (infilling) or take the left-most context from the first variable name (left-to-right mode). In the infilling setting, given that we generate the number of masks equivalent to the number of times a variable is seen, we select the most common prediction as our singular prediction. Furthermore, we only evaluate the set of variable names containing four or more characters. For our re-ranking, we consider a candidate set of 25 variables. We present our results in Table 4. We again see substantial benefits from using both left and right context: left-to-right reranking and causal-masked infilling both outperform the left-to-right single baseline (which uses only the left context). Causal-masked

---

that human-annotated type hints can be inaccurate, and that exact match does not reason about type unification or equivalence (e.g., there is no partial credit given for predicting `Optional[str]` rather than `Union[None, str]`).

[8]Our model makes a prediction for every example, so it has identical precision, recall, and F1 scores.

| Method | Accuracy |
|---|---|
| Left-to-right single | 18.4 |
| Left-to-right reranking | 23.5 |
| Causal-masked infilling | 30.6 |

Table 4: Results on the variable renaming benchmark that we construct from CodeXGLUE. Our model benefits from using the right-sided context in selecting (L-R reranking and CM infilling) and proposing (CM infilling) variable names.

infilling substantially on the left-to-right reranking method, demonstrating the value of conditioning on the right context when proposing candidate completions.

## 5 ABLATION EXPERIMENTS

For an analysis of the effects of training a model with causal masking (rather than the standard language modeling objective, as well as model size and the training data, we train several variations of our model. We compare model pass@1 scores on the HumanEval (Chen et al., 2021a) and MBPP (Austin et al., 2021) left-to-right synthesis benchmarks, with results in Table 5.

**Objective.**   Comparing 1.3B parameter models trained on the same training data with the causal masked (CM) objective (row 2) and the standard left-to-right language modeling (LM) objective (row 3), we see that the causal-masked model obtains slightly higher performance on the HumanEval and MBPP tasks in pass@1 score. This provides further evidence that causal masking training does not hurt the model's ability to perform standard left-to-right generation, at least to the 1.3B parameter scale, in line with the findings of Bavarian et al. (2022).

**Model size.**   With data fixed, increasing model size consistently improves performance (comparing the 6.7B and 1.3B CM models in rows 1 and 2, and the 1.3B and 2.3B LM models in rows 3 and 6).

**Effects of data.**   We compare models trained on our entire dataset of multiple code languages and StackOverflow (multi lang + SO, described in Section A.1) to data ablations that train only on Python code files and StackOverflow (Python + SO) and only Python code files (Python). We find that training on multiple languages gives a slight reduction in performance on these Python evaluations. However, comparing rows 4 and 5, we see that including StackOverflow data in training substantially improves performance on both HumanEval and MBPP. This suggests that future work on generative code models for language-guided synthesis tasks should consider using StackOverflow or other corpora that mix natural language and code as training data.

## 6 QUALITATIVE EXAMPLES

We show a variety of qualitative examples from our model in Section D.2 in both the infilling and left-to-right generation modes: docstring generation, metadata conditioning, class attribute inference from class usage, comment-conditioned code editing, StackOverflow title and tag generation, and zero-shot bidirectional translation of technical jargon between Chinese and English.

| # | Size (B) | Obj. | Training Data | Data Size | Train Tokens | Train Compute | HumanEval Pass@1 | MBPP Pass@1 |
|---|---|---|---|---|---|---|---|---|
| 1) | 6.7 | CM | multi lang + SO | 204 GB | 52 B | 3.0 Z | 15 | 19.4 |
| 2) | 1.3 | CM | multi lang + SO | 204 GB | 52 B | 0.6 Z | 8 | 10.9 |
| 3) | 1.3 | LM | multi lang + SO | 204 GB | 52 B | 0.6 Z | 6 | 8.9 |
| 4) | 1.3 | LM | Python + SO | 104 GB | 25 B | 0.3 Z | 9 | 9.8 |
| 5) | 1.3 | LM | Python | 49 GB | 11 B | 0.1 Z | 5 | 6.1 |
| 6) | 2.3 | LM | multi lang + SO | 204 GB | 52 B | 1.1 Z | 9 | 12.7 |

Table 5: Ablation results, comparing model performance on the Python portion of a validation set held out from our training corpora as well as the HumanEval and MBPP benchmarks. We compare models by size (in billions of parameters), objective (causal masked, CM, versus standard left-to-right language modeling, LM), training data, and total amount of compute in training (in zettaflops).

## 7 RELATED WORK

**Language Models for Code**   There has been a flurry of recent work on training large-scale neural language models on source code. Existing models differ in their architectural design and training objectives, e.g., decoder-only language models (Austin et al., 2021; Chen et al., 2021a; Izadi et al., 2022; Xu et al., 2022; Nijkamp et al., 2022), encoder-only masked language models (Feng et al., 2020; Kanade et al., 2020), and encoder-decoder models (Ahmad et al., 2021; Li et al., 2022; Roziere et al., 2021; Wang et al., 2021). Decoder-only language models have grown in popularity as they can perform zero-shot program synthesis by generating in a left-to-right fashion. On the other hand, InCoder is a decoder-only causally-masked language model that can infill arbitrary spans of text. This allows the model to perform program synthesis and many other code infilling tasks.

**Infilling Models**   Many real-world applications require infilling sequences using left and right context, e.g., editing sentences (Shih et al., 2019), restoring ancient text (Assael et al., 2019), and fixing bugs in source code. Unfortunately, standard left-to-right language models cannot directly infill text, and popular masked language models are mainly trained to infill very short spans (Chan et al., 2019; Devlin et al., 2019; Raffel et al., 2020; Roziere et al., 2021). Recent work addresses this by changing model architectures, inference procedures, and training objectives (Aghajanyan et al., 2022a; Stern et al., 2019; West et al., 2021; Aghajanyan et al., 2022b). Most related to our approach is the work of Donahue et al. (2020) and CM3 (Aghajanyan et al., 2022a), who train left-to-right language models to fill in masked token segments of varying lengths; and the work of Alon et al. (2020), who train an infilling-capable, AST-structured generative model of code on a smaller scale. In addition, concurrent to our work, OpenAI developed a fill-in-the-middle (FIM) training objective similar to the causal masking objective we use, trained code models with it, and evaluated on the HumanEval infilling tasks we introduce here (Bavarian et al., 2022). Similar to our findings in Section 5, they find that the infilling capability does not adversely affect left-to-right performance. Our objective, in contrast, allows infilling multiple regions of code, and we demonstrate the benefits of infilling across a broader range of natural programming tasks.

**Machine Learning for Code Assistance**   There is an extensive literature on using machine learning models to aid human programmers. This includes methods to infer variable types (Pradel et al., 2020; Wei et al., 2020), generate unit tests (Fraser & Arcuri, 2011), repair programs  (Gupta et al., 2017; Yasunaga & Liang, 2020; Chen et al., 2021c; Yasunaga & Liang, 2021), and verify program correctness (Ryan et al., 2020). Our model can infill arbitrary spans of code, allowing it to complete many of these tasks, as well as perform standard left-to-right generation, in a single approach.

**Machine Learning for Program Synthesis**   Program synthesis approaches directly generate programs from a specification of functionality (Gulwani et al., 2017). Such models work by taking e.g., input-output examples (Balog et al., 2017; Gulwani, 2011; Chen et al., 2021b; Bavishi et al., 2019), partial implementations (Solar-Lezama et al., 2006), or natural language descriptions (Zelle & Mooney, 1996; Yu et al., 2018; Yin et al., 2018; Kulal et al., 2019; Chen et al., 2021a) of the desired program as input. Our InCoder model differs from this past work as it can both synthesize and infill arbitrary spans of code, conditioning on natural language and partial implementations.

## 8 CONCLUSION

We demonstrated that using a causal masking objective when training a generative model of code enables strong zero-shot performance on many challenging and practical code infilling and editing tasks. The model's additional infilling capability does not appear to harm its ability to do standard left-to-right generation: ablation and comparison experiments show that our causal-masked models have comparable performance to similarly-resourced models on standard left-to-right language-to-code synthesis benchmarks. Looking forward, we expect our model performance to continue to increase with more parameters, data, and training steps (Kaplan et al., 2020; Henighan et al., 2020). Moreover, fine-tuning would allow our models to be better able to condition on natural language instructions and other indications of human intent (Zhong et al., 2021; Wei et al., 2022; Ouyang et al., 2022). Finally, our model lays a foundation for future work on supervised infilling & editing via model fine-tuning, as well as performing iterative decoding, where the model can be used to refine its own output (Ghazvininejad et al., 2019).

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

## A  DATA

### A.1  CODE DATA

**Sources.**  We obtained code files and repository metadata from GitHub and GitLab through the sites' public APIs over a period ending on December 9th, 2021. We obtained approximately 670,000 public non-fork repositories which GitHub/GitLab detected as containing primarily Python, JavaScript, or Jupyter Notebook files, and with either an MIT, Apache 2.0, BSD-2, or BSD-3 clause license. We included all code from a list of 28 languages (determined by file extension) contained in these repositories.[9] Since Python files can also be contained in non-majority-Python repositories, we also included all other Python and Jupyter files obtainable through the GitHub archive on BigQuery that we did not already obtain from GitHub directly.[10] We preprocess Jupyter notebooks by including all text and code (with Markdown formatting removed from text cells), with cells demarcated by XML-style tags (see Section A.3).

**Deduplication.**  Recent work has shown that deduplicating training data can improve model performance and reduce the risk of memorizing training data (Allamanis, 2019; Lee et al., 2022; Kandpal et al., 2022). Our deduplication scheme removes code files using exact match on the sequence of alphanumeric tokens in the file.[11] This removed approximately 75% of the corpus by file size (reducing from 1 TB to 250 GB) as there are numerous duplicated repositories, library dependencies included as source files, and common boilerplate code files (e.g., for Python web frameworks). We also use regular expressions to detect email addresses in the code files and replace them with a dummy address,[12] to reduce the risks of the model memorizing real email addresses or hallucinating fake ones.

**Decontamination.**  To ensure that our code generation models can be evaluated on several current code generation benchmarks, we perform data decontamination: removing overlap between our training data and the evaluation sets of these benchmarks. We remove any repositories contained in

---

[9]We include source files from C, C++, CSS, C#, Common Lisp, Dart, Forth, Go, HTML, Haskell, Java, JavaScript, Julia, Jupyter, Kotlin, Lua, Matlab, PHP, Perl, Python, R, Ruby, Rust, SQL, Scala, Shell, Swift, and TypeScript, although the great majority of files are Python and JavaScript. See Figure 3.

[10]We use `https://cloud.google.com/blog/topics/public-datasets/ github-on-bigquery-analyze-all-the-open-source-code`. We only include repositories with one of the above permissive licenses.

[11]We implement match checking using a Bloom filter Bloom (1970) whose keys are: the file extension, number of tokens in the file, and an MD5 hash Rivest (1992) of the sequence of tokens, which is highly accurate at identifying files with exactly matching token sequences.

[12]We replace detected email addresses with `removed@example.com`.

the validation and test sets of CodeSearchNet (Husain et al., 2019), as these are used to construct validation and test sets for several of the tasks in CodeXGLUE (Lu et al., 2021).[13]

**Filtering.**  Our filtering is similar to past work on generative models of code Chen et al. (2021a); Nijkamp et al. (2022); Xu et al. (2022): we remove files that contain any line longer than 3000 tokens or an average line length greater than 100 tokens, have less than 40% of their characters being alphanumeric or underscores, or appear to be automatically generated, which we determine using substring match on a small number of phrases produced by automatic code and documentation generation systems.[14]  Our decontamination and filtering steps together remove roughly 10% of Python files.

## A.2  STACKOVERFLOW

The second component of our corpus consists of questions, answers, and comments from Stack-Overflow. The Pile (Gao et al., 2020), which was used to train recent generative code models that we compare to in Section 5, also contains these questions and answers but does not contain comments. We include all questions that have at least one answer, up to ten answers with a non-negative score (sorted by score) per question, and up to five comments per question/answer. Qualitatively, we find that comments, together with the infilling ability of the model, allow our model to have some capability to do interactive code editing guided by language (see Figure 11).

## A.3  METADATA

We include some metadata on the code files and StackOverflow questions/answers directly in our training data to allow attribute-conditioned generation (Keskar et al., 2019; Zellers et al., 2019) and attribute prediction. For code file data, our attributes are the code filename, the file extension (as a proxy for language), the file source (GitHub or GitLab), and, for GitHub repositories, the number of stars binned into six buckets.[15]  To allow this metadata to be optional when performing left-to-right prompting of the model, we insert each attribute it the beginning of its document with a probability of 50% (allowing the model to learn metadata conditioning); otherwise, we insert it at the end of its document (allowing metadata prediction). See Figure 6a and Figure 6b for examples. For StackOverflow, our metadata attributes are the question tags for the topic (e.g., `python,django`) and the number of votes for each question and answer, binned in the same way as repository stars. We insert comments directly after the questions or answers they were written for. See Figure 6c for examples.

## A.4  TOKENIZATION

To increase the amount of context that our code model can condition on, the length of documents that the model can generate, and the efficiency of training and inference, we train a byte-level BPE tokenizer Sennrich et al. (2016); Radford et al. (2019). We allow tokens to extend across whitespace (excluding newline characters) so that common code idioms (e.g., `import numpy as np`) are represented as single tokens in the vocabulary. This substantially improves the tokenizer's efficiency— reducing the total number of tokens required to encode our training corpus by 45% relative to the byte-level BPE tokenizer and vocabulary of GPT-2.

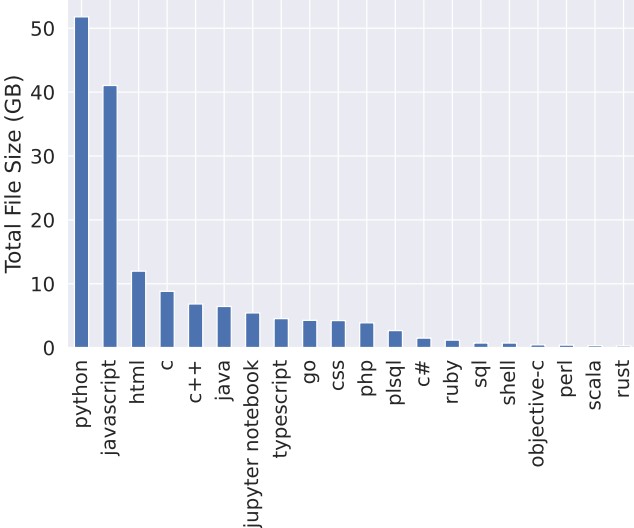

Figure 3: Code corpus composition (after deduplication and filtering) by total file size for the most common languages, as determined by file extension.

## A.5 CORPUS STATISTICS

See Figure 3 for a plot showing code corpus composition (after deduplication and filtering) by total file size for the most common languages, as determined by file extension.

## B MODEL AND INFERENCE DETAILS

### B.1 MODEL

| Parameter | INCODER-1.3B | INCODER-6.7B |
|---|---|---|
| –decoder-embed-dim | 2048 | 4096 |
| –decoder-output-dim | 2048 | 4096 |
| –decoder-input-dim | 2048 | 4096 |
| –decoder-ffn-embed-dim | 8192 | 16384 |
| –decoder-layers | 24 | 32 |
| –decoder-normalize-before | True | True |
| –decoder-attention-heads | 32 | 32 |
| –share-decoder-input-output-embed | True | True |
| –decoder-learned-pos | False | False |

Table 6: Fairseq architecture hyperparameters for our INCODER models.

Our primary model is INCODER-6.7B, a 6.7B Transformer Vaswani et al. (2017) language model. We use the same architecture as the dense 6.7B models described in Artetxe et al. (2021); the Fairseq architecture description can be found in Table 6. INCODER-6.7B was trained on 248 V100 GPUs for

[13]We also search for occurrences of code from the HumanEval Chen et al. (2021a) dataset using substring match, but we did not find any matches in our training set. The solutions to the problems in this dataset, at the time that we obtained our files from GitHub, were only contained in JSON files. We additionally removed repositories in the evaluation sets of the JuICe tasks (Agashe et al., 2019), although we did not evaluate our model on these tasks in this present work.

[14]Our filters target files automatically generated by the Protocol Buffer Compiler, Django, Epydoc, Pdoc, Sphinx, MkDocs, or MathJax.

[15]We size bins using an inverse binary logarithmic scale, so that bucket 0 corresponds to the repositories with star counts up to the 50th percentile, bucket 1 corresponds to the 50th to 75th percentiles, and bucket 5 to those in the 97th percentile and above.

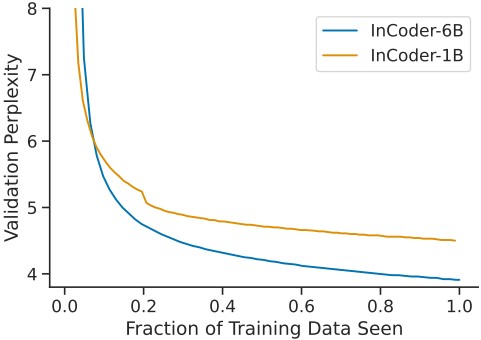
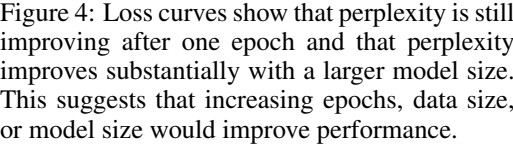
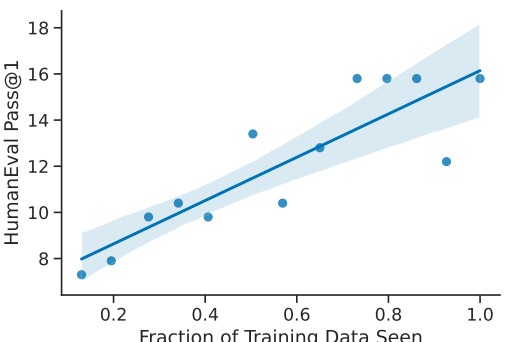

Figure 4: Loss curves show that perplexity is still improving after one epoch and that perplexity improves substantially with a larger model size. This suggests that increasing epochs, data size, or model size would improve performance.

Figure 5: Performance of INCODER-6.7B on the HumanEval left-to-right synthesis benchmark generally increases over the course of pre-training. We plot a line of best fit along with a 95% confidence interval via bootstrap resampling.

24 days. We perform one epoch on the training data, using each training document exactly once. Our implementation utilized the causal masking implementation (Aghajanyan et al., 2022a) available in Fairseq (Ott et al., 2019), with the underlying library being PyTorch (Paszke et al., 2019). Our per-GPU batch size was 8, with a maximum token sequence length of 2048. We clip all gradient norms to 1.0 and used the Adam optimizer with $\beta_1 = 0.9$, $\beta_2 = 0.98$ (Kingma & Ba, 2015). For our learning rate scheduler, we use the built-in polynomial decay learning rate scheduler available in Paszke et al. (2019) with 1500 warmup updates. Fairscale was used for improving memory efficiency through fully sharding model states (Baines et al., 2021).

We compare the validation perplexity of the 6B parameter model and a smaller 1.3B parameter model (see Section 5 for details on the training of this 1.3B model) in Figure 4, showing comparable scaling laws to those reported by Aghajanyan et al. Aghajanyan et al. (2022a). Our models have also not yet saturated and would benefit from further training; we report the performance of the 6.7B model on the HumanEval Python function synthesis benchmark (Chen et al., 2021a) (see Section C.6 for a description of this benchmark) and see a consistent increase in performance over the course of training (Figure 5).

## B.2 INFERENCE DETAILS

In practice, to generate a single infill we sample from the distribution $P(\cdot \mid$ [Left; <Mask:0>; Right; <Mask:1>; <Mask:0>]), where we insert an artificial <Mask:1> token. Not inserting <Mask:1> gives an implicit size hint to the model that the <Mask:0> token should be expanded to fill the rest of the 2048 token context window. Instead, inserting a <Mask:1> token indicates to the model that some amount of the document is omitted after the right context. We found that including this substantially improved the ability of the model to predict <EOM> appropriately when generating an infill for <Mask:0>. See Aghajanyan et al. (2022a) for more.

More generally, when inserting at multiple locations, we condition on the document with multiple mask sentinel tokens inserted and a final mask token appended. For example, to insert at two locations we use [A; <Mask:0>; C; <Mask:1>; E; <Mask:2>]) and infill the masks in order, appending the appropriate <Mask:k> sentinel tokens to signal the start of generation for the next span, i.e., the completed document for two insertion locations is represented by [A; <Mask:0>; C; <Mask:1>; E; <Mask:2>; <Mask:0>; B; <EOM>; <Mask:1>; D; <EOM>], where regions B and D have been infilled.

## C EXPERIMENTAL DETAILS AND SUPPLEMENTARY RESULTS

### C.1 INFILLING COMPARISONS

We describe our adaptation of models from prior work to the zero-shot infilling setting, for the experiments described in Section 4.

**Encoder-decoder (PLBART).** We use PLBART-Large (Ahmad et al., 2021), an encoder-decoder model trained on code (including 220GB of Python) using a BART (Lewis et al., 2019) masked denoising objective. We pre- and post-process each HumanEval infilling example as needed for PLBART: we represent each example as a stream of space-separated tokens (as identified by Python's built-in lexer) with newlines and indentations replaced by control characters, and use a <mask> token to represent the line to be infilled. We extract the infilled region from the output by searching for the longest suffix of the left context contained in the output, and (as in our left-to-right baselines) take the ground-truth number of lines following this left context suffix as the infill.

**Left-to-right with templated prompting (Codex).** We perform zero-shot prompting on the Codex code-cushman-001 and code-davinci-001 OpenAI API models using the following template:

```
[code before the infill mask] <INFILL> [code after the infill mask]
# Complete the above code by replacing the <INFILL> tag.
[code before the infill mask]
```

We take [code after the infill mask] as the indicator of completion.

### C.2 CODE CLOZE (CODEXGLUE)

CodeXGLUE cloze is created from CodeSearchNet to evaluate CodeBERT and consists of a short natural language description followed by code in several programming languages. We evaluate on the max/min subtask, where the model has to decide if the given mask should be filled with either max or min. Since there are only two options in this task, we can closely compare the causal-masked infilling and left-to-right setups by scoring both options and selecting the sequence with the highest likelihood.

Table 7 contains the main results. Using the causal-masked infill format with a single token (containing min/max) as the masked region (CM infill-token) performs better than using just the left context, but not as well as scoring the entire sequence left to right. Masking a larger region (CM infill-region), containing the left prefix and 10 right-side tokens in the masked region, performs comparably to scoring the whole sequence. Infill region length and tokenization can affect the performance, see C.3 for more details and more comparisons.

| Method | Python | JavaScript | Ruby | Go | Java | PHP |
|---|---|---|---|---|---|---|
| Left-to-right single | 76.9 | 77.6 | 65.8 | 70.4 | 74.1 | 77.1 |
| Left-to-right reranking | **87.9** | 90.1 | 76.3 | 92.8 | **91.7** | 90.4 |
| CM infill-token | 81.8 | 73.9 | 81.6 | **95.4** | 77.6 | 87.0 |
| CM infill-region | 86.2 | **91.2** | 78.9 | 94.7 | 89.8 | **91.4** |
| CodeBERT | 82.2 | 86.4 | **86.8** | 90.8 | 90.5 | 88.2 |

Table 7: Accuracy on the CodeXGLUE max/min cloze task. We compare four different inference methods. Left-to-right single: scoring with left-to-right ordering using only the left context and the completion (containing max or min); Left-to-right reranking: scoring with left-to-right ordering using the left context, completion, and right context; CM infill-token: causal masking scoring, using only a single token (containing max or min) as the infill, CM infill-region: causal masking scoring that additionally contains 10 tokens from the right side context.

Note that comparing the scores of the sequences, which differ in their infills, with the left-to-right setup is more computationally expensive than with the CM infilling setup, as the Transformer intermediate activations can be cached and shared across identical sequence prefixes, and in the CM infill setup all sequence differences occur at the ends.

## C.3    CLOZE AND SINGLE TOKEN INFILL DETAILS

|  | Python | Javascript | Ruby | Go | Java | PHP |
|---|---|---|---|---|---|---|
| Left-break | 72.4 | 72.1 | 68.4 | 71.7 | 74.1 | 76.9 |
| Left-token | 76.9 | 77.6 | 65.8 | 70.4 | 74.1 | 77.1 |
| Left-region | 84.2 | 88.6 | 73.7 | 85.5 | 87.6 | 87.0 |
| Left-right-break | 77.9 | 79.4 | 63.2 | 89.5 | 82.0 | 85.3 |
| Left-right | 87.9 | 90.1 | 76.3 | 92.8 | 91.7 | 90.4 |
| Infill-break | 79.1 | 83.1 | 84.2 | 90.1 | 84.0 | 85.3 |
| Infill-token | 81.8 | 73.9 | 81.6 | 95.4 | 77.6 | 87.0 |
| Infill-region | 86.2 | 91.2 | 78.9 | 94.7 | 89.8 | 91.4 |
| CodeBERT | 82.2 | 86.4 | 86.8 | 90.8 | 90.5 | 88.2 |
| Codex[*] | 93.6 | 93.4 | 94.7 | 99.3 | 95.0 | 94.3 |

Table 8: Accuracy on CodeXGLUE cloze max/min. Left: scoring using only the left context, Left-right: score the whole program, Infill: score the infilling sequence, -region: include left context and 10 tokens from the right. -break: break tokenization on the infilled token. Codex[*]: version `code-davinci-001` of OpenAI's Codex model, as accessed through their API. Information on the training data for this model is unclear, and it may contain portions of CodeSearchNet (which contains this task's evaluation set).

As shown in Table 8, breaking tokenization (-break) on infill decreases the performance using all scoring methods. For example, whereas `Math.max(` was a single token in the full sequence, the sequence is broken into `Math.`, `max`, and `(` for infilling. Infilling with the original tokenization increases the performance slightly, but does not match full left-right scoring. We suspect this is because the model was not trained on infilling single tokens, unlike CodeBERT. A way to fix this is to include a larger region on the left and a few more tokens on the right. This will only slightly increase the scoring complexity. To show that our model uses the right context, we compare it with scoring the left-only model. More precisely, the sequences being scored are

```
       Left-to-right single:  [Left; Token]
  Left-to-right reranking:  [Left; Token; Right]
             Infill-token:  [Left; <Mask:0>; Right; <Mask:1>; <Mask:0>; Token; <EOSS>]
              Left-region:  [Left; Token; Right[:10]]
            Infill-region:  [ <Mask:0>; Right[10:]; <Mask:1>; <Mask:0>; Left; Token; Right[:10]; <EOSS>]
```

## C.4    COMPARISON TO OPENAI'S CODE API

We evaluate OpenAI's proprietary `code-davinci-002` system, accessed through their API, on our single-line HumanEval infilling task, with results given in Table 9, There is limited public information about this system, including on its training data or procedure (although Bavarian et al. 2022 describes their FIM objective as early research that helps power the model), how it performs infills, or whether any postprocessing is done on model outputs, but we report its performance to help gauge the difficulty of our new task. For both `code-davinci-002` and our INCODER-6.7B model, conditioning on right-sided context improves performance, with the most substantial improvements from infilling.

| Model | Inference | Pass Rate | Exact Match |
|---|---|---|---|
| INCODER-6.7B | Left-to-right single | 48.2 | 38.7 |
| INCODER-6.7B | Left-to-right reranking | 54.9 | 44.1 |
| INCODER-6.7B | Infilling | 69.0 | 56.3 |
| code-davinci-002 | Left-to-right single | 63.7 | 48.4 |
| code-davinci-002 | Left-to-right reranking | 71.8 | 52.0 |
| code-davinci-002 | Infilling | 87.4 | 69.6 |

Table 9: We evaluate OpenAI's proprietary code-davinci-002 system, accessed through their API, on our single-line HumanEval infilling task. Although no information is currently public about this system, its training data or procedure, how it performs infills, or whether any postprocessing is done on model outputs, we report its performance to help gauge the difficulty of our new task. For both code-davinci-002 and our INCODER-6.7B model, conditioning on right-sided context improves performance, with the most substantial improvements from infilling.

## C.5 ADDITIONAL TYPE HINT PREDICTION SETTING

Our results in Section 4.3 filtered out functions from the TypeWriter prediction set which had a return type hint of None, as these type hints are are overrepresented in the dataset in compared to naturally-occurring code, due to the filtering process used to construct it. For a closer comparison to the setting used in the original TypeWriter paper, we present results including these functions in Table 10. Given TypeWriter's static analysis capabilities, and the overrepresentation of None types in this evaluation set, we add a simple post-processing step (*return checks*) that predicts None if the function does not have any non-trivial return statements, which captures some of the effect of TypeWriter's analysis capabilities. In all settings, our zero-shot infill approach outperforms the left-to-right baselines, and obtains performance comparable to the supervised TypeWriter approach when return checks are used.

| Method | Precision | Recall | F1 |
|---|---|---|---|
| Ours: Left-to-right single | 20.0 | 20.0 | 20.0 |
| Ours: Left-to-right rerank | 24.2 | 24.2 | 24.2 |
| Ours: Infill | 46.8 | 46.8 | 46.8 |
| Ours: Left-to-right single + Return checks | 63.2 | 63.2 | 63.2 |
| Ours: Left-to-right rerank + Return checks | 64.3 | 64.3 | 64.3 |
| Ours: Infill + Return checks | 76.7 | 76.7 | 76.7 |
| TypeWriter (Supervised) | 78.8 | 69.9 | 74.1 |

Table 10: Return type hint prediction results on the 25% subset of TypeWriter's OSS dataset where were able to obtain source files, extract functions and types from, and that were not contained in our model's training set. Given an overrepresentation of functions with None in this dataset, and the static analysis capabilities of TypeWriter, we also give results using a simple post-processing step that predicts None if the function does not have any non-trivial return statements.

## C.6 COMPARISON TO LEFT-TO-RIGHT GENERATIVE MODELS ON CODE SYNTHESIS

We compare to past published work on generative code models on the HumanEval (Chen et al., 2021a) and MBPP (Austin et al., 2021) benchmarks, which require models to condition on natural language descriptions (docstrings) to produce Python programs (typically a single function), and evaluates overall functional accuracy (pass rate) across examples using several test cases for each program.

We evaluate our INCODER-6.7B model in zero-shot evaluation on both of these benchmarks. For HumanEval, we follow past work by prompting with function signatures and docstring descriptions, sample 200 candidate program completions, and compute pass@1, pass@10, and pass@100 using the unbiased sampling estimator of Chen et al. (Chen et al., 2021a). For MBPP, which

| Model | Size (B) | Python Code (GB) | Other Code (GB) | Other (GB) | Code License | Infill? | HE @1 | HE @10 | HE @100 | MBPP @1 |
|---|---|---|---|---|---|---|---|---|---|---|
| *Released* | | | | | | | | | | |
| CodeParrot (Tunstall et al., 2022) | 1.5 | 50 | None | None | — | | 4.0 | 8.7 | 17.9 | — |
| PolyCoder (Xu et al., 2022) | 2.7 | 16 | 238 | None | — | | 5.6 | 9.8 | 17.7 | — |
| GPT-J (Wang & Komatsuzaki, 2021; Chen et al., 2021a) | 6 | 6 | 90 | 730 | — | | 11.6 | 15.7 | 27.7 | — |
| INCODER-6.7B | 6.7 | 52 | 107 | 57 | Permissive | ✓ | 15.2 | 27.8 | 47.0 | 19.4 |
| GPT-NeoX (Black et al., 2022) | 20 | 6 | 90 | 730 | — | | 15.4 | 25.6 | 41.2 | — |
| CodeGen-Multi (Nijkamp et al., 2022) | 6.1 | 62 | 375 | 1200 | — | | 18.2 | 28.7 | 44.9 | — |
| CodeGen-Mono (Nijkamp et al., 2022) | 6.1 | 279 | 375 | 1200 | — | | 26.1 | 42.3 | 65.8 | — |
| CodeGen-Mono (Nijkamp et al., 2022) | 16.1 | 279 | 375 | 1200 | — | | 29.3 | 49.9 | 75.0 | — |
| *Unreleased* | | | | | | | | | | |
| LaMDA (Austin et al., 2021; Thoppilan et al., 2022; Chowdhery et al., 2022) | 137 | None | None | ??? | — | | 14.0 | — | 47.3 | 14.8 |
| AlphaCode (Li et al., 2022) | 1.1 | 54 | 660 | None | — | | 17.1 | 28.2 | 45.3 | — |
| Codex-2.5B (Chen et al., 2021a) | 2.5 | 180 | None | > 570 | — | | 21.4 | 35.4 | 59.5 | — |
| Codex-12B (Chen et al., 2021a) | 12 | 180 | None | > 570 | — | | 28.8 | 46.8 | 72.3 | — |
| PaLM-Coder (Chowdhery et al., 2022) | 540 | ~20 | ~200 | ~4000 | Permissive | | 36.0 | — | 88.4 | 47.0 |

Table 11: A comparison of our INCODER-6.7B model to published code generation systems using pass rates @ $K$ candidates sampled on the HumanEval and MBPP benchmarks. All models are decoder-only transformer models. A "Permissive" code license indicates models trained on only open-source repositories with non-copyleft licenses. The GPT-J, GPT-NeoX, and CodeGen models are pre-trained on The Pile (Gao et al., 2020), which contains a portion of GitHub code without any license filtering, including 6 GB of Python. Although the LaMDA model does not train on code repositories, its training corpus includes ~18 B tokens of code from web documents. The total file size of the LaMDA corpus was not reported, but it contains 2.8 T tokens total. We estimate the corpus size for PaLM using the reported size of the code data and the token counts per section of the corpus.

does not include function signatures, we prompt only with the docstring description and compute pass@1 (Chowdhery et al., 2022) using a single candidate.[16] We use top-p sampling with $p = 0.95$, with a temperature of 0.2 for pass@1 and 0.8 for pass@10 and pass@100.

We compare our INCODER-6.7B model to models from past work (which have all been left-to-right only) in Table 11, giving the model size and training data summary statistics as reported (or estimated, in cases when a paper only reports token counts, as tokenizer efficiencies vary) in these papers.While differences in details of the Transformer model architectures, datasets, and training procedures across papers and experimental setups make a rigorous comparison impossible, we note that our model achieves roughly comparable performance on the HumanEval metrics to CodeGen-Multi (Nijkamp et al., 2022), which is also a ~6B parameter model trained on roughly the same amount of Python code, as well as AlphaCode's 1.1B decoder-only model (Li et al., 2022) which also uses a similar amount of Python training data.

---

[16]While this setting is not directly comparable to the three-shot setting where the models of Austin et al. (Austin et al., 2021) and Chowdhery et al. (Chowdhery et al., 2022) performed best, we found that our model did not benefit from additional examples in the prompt, which we attribute to much smaller size of our model (6.7B, versus 137B or 540B parameters) and the sensitivity of in-context learning to model scale.

# D    EXAMPLES

## D.1    METADATA EXAMPLES

```
<| file ext=.py source=github |>
from typing import Dict

def count_words(filename: str) -> Dict[str, int]:
    """Count the number of occurrences of each word in the file."""
    with open(filename, 'r') as f:
        word_counts = {}
        for line in f:
            for word in line.split():
                if word in word_counts:
                    word_counts[word] += 1
                else:
                    word_counts[word] = 1
    return word_counts
<|/ file filename=string_utils.py dstars=4>
```

(a) Metadata for code includes the file extension, source (github or gitlab), filename, and binned number of stars for GitHub repositories (in logarithmically-sized bins numbered 0 to 5, see Section A.3).

```
<| file ext=.ipynb:python |>
<text>
This notebook demonstrates using scikit-learn to perform PCA.
</text>
<cell>
%matplotlib inline
</cell>
<cell>
import numpy as np
import matplotlib.pyplot as plt
from sklearn.decomposition import PCA
</cell>
...
```

(b) In addition to the standard metadata used for all other code files, Jupyter Notebook metadata includes the kernel type (in this instance, Python) as well as the type of the cells in the notebook (either code or text).

```
<| q tags=python,django |>
Django - How to display a form with user input

I'm trying to create a simple web application that can
allow a user to enter a username and password.
Here's what I have so far:

from django.shortcuts import render
...

Can you help?
<|/ q dscore=3 |>
<| c |>
Why aren't you using the django authentication library?
<|/ c |>
<| a dscore=4 |>
You need to add your form to the template:

...

<| a tags=python,django |>
```

(c) Metadata attributes for StackOverflow include question tags and discretized scores of questions and answers.

Figure 6: Examples of metadata attributes included in the training data to allow attribute-conditioned generation and attribute prediction. To allow both generation and prediction, attributes are randomly included either at the beginning of the document or at the end (with probability 0.5 each). Attributes occur in random order to allow arbitrary orderings at inference time.

## D.2 EXAMPLE MODEL OUTPUTS

```python
def get_conn(self):
    """ """

    conn_config = self._get_conn_params()
    conn = snowflake.connector.connect(**conn_config)
    return conn
```

(a) Reference docstring: `Returns a snowflake.connection object.`
Model docstring: `Establishes a connection to the Snowflake cluster.`

```python
def sprint(text, *colors):
    """ """

    return "\33[{}m{content}\33[{}m".format(";".join([str(color) \
        for color in colors, RESET, context=text])) \
        if IS_ANSI_TERMINAL and colors else text
```

(b) Reference docstring: `Format text with color or other effects into ANSI escaped string.`
Model docstring: `Prints a string with ANSI color codes.`

Figure 7: Example docstring generations for the CodeXGLUE code-to-text dataset. Captions for each example give the reference human-written docstring and the output from our INCODER-6.7B model with causal-masked infilling. The model generates docstrings zero-shot by inserting text between the """ comment delimiters.

```
<| file ext=.py |>
# count the words in all files in the current directory
import os
import sys

def main():
  cwd = os.getcwd()

  words = 0

  for filename in os.listdir(cwd):
    if filename.endswith(".in"):
      fname = os.path.join(cwd, filename)
      with open(fname) as infile:
        for line in infile:
          words += len(line.split())

  print(words)

if __name__ == '__main__':
  main()
```

```
<| file ext=.sh |>
# count the words in all files in the current directory
find . -type f -name "*.txt" -exec wc -w {} \; | sort -nr | head -n 20
```

Figure 8: Meta-data conditioning on file extensions for Python (left) and Shell (right) allows completing the same text comment as either a Python script or a pipelined bash command, respectively. Regions highlighted in orange are left-to-right generations from our INCODER-6.7B model.

```
class Person:

    def __init__(self, name, age, gender):
        self.name=name
        self.age=age
        self.gender=gender

p = Person('Eren', 18, "Male")
```

Figure 9: Given the beginning of a class definition and right-sided context of the class being used, the model is able to infer plausible attribute names for the class (e.g., "Eren" is likely to be a name, 18 is age, "Male" is the gender.) The region highlighted in orange is an infill generation from our INCODER-6.7B model.

**Original Code**

```
data, target = data.to(device), target.to(device)
optimizer.zero_grad()
output = model(data)
loss = F.nll_loss(output, target)
loss.backward()

data, target = data.to(device), target.to(device)
optimizer.zero_grad()
output = model(data)
loss = F.nll_loss(output, target)
loss.backward()
```

**1. Insert comment and regenenerate**

```
data, target = data.to(device), target.to(device)
optimizer.zero_grad()
output = model(data)
loss = F.nll_loss(output, target)
loss.backward()

data, target = data.to(device), target.to(device)
optimizer.zero_grad()
output = model(data)
# use a Huber loss
criterion = nn.SmoothL1Loss()
loss = criterion(output, target)
loss.backward()
```

Figure 10: By inserting a comment (# use a Huber loss) in the code and replacing the line after with a mask token, a user can prompt the model to in-fill a region of code with a contextually-appropriate replacement. Lines in orange are in-fill generations from our INCODER-6.7B model.

**1. Generate answer conditioned on question**

```
I need to write a Python function called `snake_to_camel`. Here's a description:
"Write a function to convert a snake case string to a camel case string."
<| q tags=python |>
<| a |>
You can use str.title:

>>> 'my_snake_case'.title()
'MySnakeCase'

</ a tags=python dscore=3 |>
```

**2. Add comment**

```
I need to write a Python function called `snake_to_camel`. Here's a description:
"Write a function to convert a snake case string to a camel case string."
<| q tags=python |>
<| a |>
You can use str.title:

>>> 'my_snake_case'.title()
'MySnakeCase'

</ a tags=python dscore=3 |>
<| c |>
Close, but you need to replace the underscores with an empty string.
<|/ c |>
<| c |>
Thanks, edited!
<|/ c |>
```

**3. Regenerate infill conditioned on comment**

```
I need to write a Python function called `snake_to_camel`. Here's a description:
"Write a function to convert a snake case string to a camel case string."
<| q tags=python |>
<| a |>

def snake_to_camel(name):
  return ''.join(x.title() for x in name.split('_'))

</ a tags=python dscore=3 |>
<| c |>
Close, but you need to replace the underscores with an empty string.
<|/ c |>
<| c |>
Thanks, edited!
<|/ c |>
```

Figure 11: Pretraining on StackOverflow, and our model's infilling capability, allows it to perform zero-shot interactive refinement of a function. In the first example, the model has generated the orange region conditioned on the user description. In the second example, the user has added a comment specifying a refinement to the function, and selected the text for the model to replace. In the third example, the orange region has been infilled by the model.

```
<| q tags=tensorflow, gpu |>
How to limit GPU memory usage in TensorFlow?

I am interested in limiting the GPU memory usage in TensorFlow.
I work in an environment in which computational resources are
shared, i.e., we have a few server machines equipped with a few
Nvidia Titan X GPUs each.
For small to moderate size models, the 12 GB of the Titan X is
usually enough for 2-3 people to run training concurrently on
the same GPU. If the models are small enough that a single model
does not take full advantage of all the computational units of
the GPU, this can actually result in a speedup compared with
running one training process after the other. Even in cases
where the concurrent access to the GPU does slow down the
individual training time, it is still nice to have the
flexibility of having multiple users simultaneously train on the
GPU.
The problem with TensorFlow is that, by default, it allocates
the full amount of available GPU memory when it is launched.
Even for a small two-layer neural network, I see that all 12 GB
of the GPU memory is used up.
Is there a way to make TensorFlow only allocate, say, 4 GB of
GPU memory, if one knows that this is enough for a given model?
<|/ q dscore=3 |>
```

Figure 12: Question tag and title prediction from the text of a StackOverflow question. Regions highlighted in orange are infill generations from our INCODER-6.7B model.

```
冒泡排序法 -> bubble sort
动态规划法 -> dynamic programming
快速排序法 -> quick sort
数据结构 -> data structure
词嵌入 -> word embedding
```

Figure 13: Zero-shot bidirectional technical jargon translation between Chinese and English. Regions in orange are infill generations from our INCODER-6.7B model.

