# OpenReview forum: "InCoder: A Generative Model for Code Infilling and Synthesis"
_ICLR.cc/2023/Conference — ICLR 2023 notable top 25%_

### Official Review · Reviewer_CKVi · 2022-10-20

**Confidence:** 5
**Correctness:** 3
**Technical Novelty And Significance:** 2
**Empirical Novelty And Significance:** 3
**Recommendation:** 6

**Clarity, Quality, Novelty And Reproducibility:**

Clarity: The approach proposed seems clearly presented. However, in the figure, I would not call these 'zero-shot' inference since the model is trained specifically to do such infilling, correct?

Quality: Missing an important comparison, as mentioned above. However, there are a lot of important evaluations (single line, multi line etc, as well as entire scope + execution based evaluation).

Novelty: Moderate, but has a good amount of impact.

Reproducibility: This work releases model weights, so it seems reproducible (even though the code release on the inference is slightly lacking).

**Strength And Weaknesses:**

Pros:

This paper shows that a simple method works.

Cons:

There is no baseline against the traditional left to right approach. That is, with a given amount of data / compute this work uses, what would be the performance if we use these resources for left to right completion alone? This is important because the amount of data that the paper uses as well as the training resources are slightly different from other papers, therefore, it would be good to see how much this amount of data/compute yield in terms of left to right completion. Note that this could be different from the left-to-right performance that is obtained from the infilling method proposed. That goal here would be to compare if this approach results in performance drop due to infilling or not, and if so how much. A similar work by Bavarian 2022 shows that with a similar approach but only with 1 infilling spot, there seems to be no degradation, but this work is sufficiently different that it is not clear if the results by Bavarian 2022 also applies here.


**Summary Of The Paper:**

The paper proposes an approach to perform code infilling that supports multiple insertion points. The approach is by defining an infilling format that describes the missing code locations and the code to be infilled. The training approach is by the traditional autoregressive next token prediction loss. This paper indeed proposes a simple technique that seems to work reasonably well.

**Summary Of The Review:**

Nice contribution, but lacking thorough comparison to the normal baseline / how much performance drops compared to normal left to right approach. However, the multiple infilling spots is a plus.

---

> ### Author Response · Authors · 2022-11-15
> **Response to Reviewer CKVi**
>
> Thank you for your review! We respond to your point here.
>
> ### Comparison to a left-to-right baseline
>
> Our ablation results in Section 5 and Table 5 compare the standard left-to-right (LM) and our loss (CM) using the same training data and same size model, and find comparable (or slightly better) performance with CM, confirming the result of Bavarian et al. We've added references to this section in the Introduction in the updated version of the paper to hopefully make these clearer.

---

> > ### Comment · Reviewer_CKVi · 2022-11-15
> > **Only row 2 and 3 are directly comparable, with some performance differences**
> >
> > In term of controlling for the same data and compute, only row 2 and 3 are comparable where the performance differences are 6.0 and 8.0 for HumanEval and 8.9 and 10.9 for HumanEval (so roughly 20% perhaps). The difference could potentially grow for larger models than 1.3 B or larger data/compute. So it does not seem like we can get this capabilities for free like in Bavarian 2022.
> >
> > I think a full ablation study on the trade off for larger sizes / higher compute would be quite important and the paper needs to be upfront about the trade-off.

---

> > > ### Author Response · Authors · 2022-11-15
> > > **Re: Only row 2 and 3 are directly comparable, with some performance differences**
> > >
> > > Thanks for your quick response and comment!
> > >
> > > Yes, only rows 2 and 3 (the 1.3B models) are comparable in Table 5. But we note that the causally-masked-trained model (row 2, CM) is actually slightly better than the standard left-to-right model (row 3, LM) on both HumanEval and MBPP. This difference could certainly shrink or grow (or change direction!) with larger model scales or larger data/compute. But, as is, this finding is currently in line with Bavarian et al. 2022 (i.e. the CM capability comes for free).
> > >
> > > We agree that it would be ideal to validate this finding at larger model sizes, but this is quite expensive computationally (and even Bavarian et al., with OpenAI's compute resources, only went to 6.9B parameters). We've updated the paper now to add some wording in Sec 5 with the caveat that our validation is limited to the 1.3B scale.

---

### Official Review · Reviewer_bKjD · 2022-10-24

**Confidence:** 4
**Correctness:** 3
**Technical Novelty And Significance:** 3
**Empirical Novelty And Significance:** 3
**Recommendation:** 6

**Clarity, Quality, Novelty And Reproducibility:**

The paper is very clear.

I think the algorithmic novelty is predictable in a sense that masking and filling mask regions have been used in other context, but it is interesting to see its results in code infilling tasks.

The paper uses custom dataset generation steps, so without exact details of those, it would be challenging to reproduce the paper.

**Strength And Weaknesses:**

The paper is well written and easy to read. The approach described tackles fundamental challenges in autoregressive generation by training  models that are capable of filling the masked regions. This ability is necessary for tasks such return type prediction and doc string generation where the output depends on both left and right context. The experimental results are also encouraging as for various tasks shown, the model is able to perform better with less model size (e.g., compared to code-davinci-001).

The paper, however, does not discuss full code generation as it focuses primarily on infilling experiments, so its competitiveness on more challenging tasks is not known. With the ability of infill and the original motivation that "Code is seldom written in a single left-to-right pass and is instead repeatedly edited and refined.", I'd be very interested to see how the model performs on full code generation as well as devising an ability to detect and re-write portions of the code when solving those tasks.

**Summary Of The Paper:**

This paper introduces a code synthesis model that is trained to perform both autoregressive left-to-right generation as well as infilling via added mask tokens. The experimental results show improved performance on various infilling related code generation tasks.

**Summary Of The Review:**

Despite not evaluating on full code generation task, the paper has enough experimental results to support itself for infilling tasks.

---

> ### Author Response · Authors · 2022-11-15
> **Response to Reviewer bKjD**
>
> Thank you for your review! We respond to your point here.
>
> ### Full code generation
>
> Our experiments on the HumanEval and MBPP benchmarks in Section 5 require the model to generate entire Python functions left-to-right, and we compare scores for our models to past work in Table 11 in the appendix. We've added references to this section in the Introduction in the updated version of the paper to hopefully make these clearer. It would be interesting to evaluate on other generation benchmarks as well (e.g. APPS), but given the prevalence of HumanEval and MBPP in the literature, we felt these were sufficient to benchmark our left-to-right generation ability in this paper.

---

### Official Review · Reviewer_7zUC · 2022-10-25

**Confidence:** 4
**Correctness:** 4
**Technical Novelty And Significance:** 3
**Empirical Novelty And Significance:** 3
**Recommendation:** 8

**Clarity, Quality, Novelty And Reproducibility:**

This paper is well-writtern and is of high quality. The proposed method is novel. And I expect that the results should be easy to reproduce.

**Strength And Weaknesses:**

Strength:
- The paper is well-written. The method is well explained with examples. Experiment results are extensive with analysis.
- The proposed method is a simple but effective masking objective. Zero-shot performance demonstrates the effectiveness for infilling line completion task, as well as other related prediction tasks.


Weakness:
- From the main text, I cannot infer whether the training is done 100% with mask infilling or there are some probability that normal left-to-right data is used for training without any masking. If it is 100% mask infilling, it seems surprising to me that left-to-right is not affected at all. I would expect normal left-to-right data exist in training, like the concurrent work fill-in-the-middle (FIM) training (Bavarian et al., 2022).
- Some training details seem missing, for example, how many training tokens and number of iterations the model has gone through during the training process.
- Though there is an <EOM> token indicating the end of generation, there is possibility that the generated code after <MASK:0> will contain overlapping content as the actual right context. Did you see any such issue exist? How do you resolve such issue?

**Summary Of The Paper:**

This paper proposes a mask infilling method to train language model for program synthesis with the ability of code infilling. The method randomly selects a span and replace it with a mask token, and place the span after the sequence as the target. The proposed method is novel, however the masking method is also used in the literature of T5 pretraining. Experiments show that the mask infilling does not degrade left-to-right performance, while significantly improve insertion capability.

**Summary Of The Review:**

Overall, I think this is a good paper with novel method proposed and well-performed experiments.

---

> ### Author Response · Authors · 2022-11-15
> **Response to Reviewer 7zUC**
>
> Thank you for your review! We respond to your questions here:
>
> ### Mask infilling training fraction
>
> While we do apply the causal masking scheme to every training example, the causal masking scheme can occasionally result in all tokens in the document occurring in their original left-to-right order: if the sampled span count is 1, and the sampled length of this span is the full document length. We theorize that these occurrences, while rare in absolute terms, still occur enough times over the course of pre-training to let the model learn the left-to-right ability. It would be interesting to repeat the FIM ablations of Bavarian et al. with much higher FIM rates (between 90% and 100%, the two high rates evaluated in their paper) in order to see what the minimal amount of left-to-right training necessary is.
>
> ### Amount of training
>
> Our training corpus consisted of 46B tokens of code, and roughly 10B tokens from StackOverflow. We performed exactly one iteration of training on this data.
>
> ### Overlap in generation after MASK:0
>
> This is a great question. We did initially see very large amounts of overlap between the infills and the right context, but this was much reduced by the sentinel insertion trick which we mention in footnote 1 and describe in Section B.2 (and which was enabled by our multi-infill training). Our evaluation tasks also afford stop tokens (e.g. """ for docstring generation) which can be used to truncate infills before they begin repeating the right context.

---

### Official Review · Reviewer_PWMt · 2022-10-25

**Confidence:** 4
**Correctness:** 3
**Technical Novelty And Significance:** 4
**Empirical Novelty And Significance:** 3
**Recommendation:** 8

**Clarity, Quality, Novelty And Reproducibility:**

Table 11 might have included CODEX-2.5B (in addition to the 12B). The performance of Codex 2.5B in HumanEval surpass InCoder, but as stated in the weaknesses section it is expected given the change in the training regime


**Strength And Weaknesses:**

Strengths
A really good paper
The idea is clever, simple and straightforward. It provides a functionality that general GPT-like or full transformer models don’t.
Experimentation seems appropriate. Authors make a great effort not sticking with standard benchmarks which might not be the ideal scenario for InCoder

Weaknesses
The training regime comes with a cost and the Left to right generation, specifically on Table 11


**Summary Of The Paper:**

The current work presents InCoder, an alternative approach to general CLM or seq-to-seq code generation where the training regime is changed to allow code generation in arbitrary subsections of the code.
Experimentation seems appropriate, including HumanEval, DocString generation, return type prediction and variable name prediction showing model capabilities.


**Summary Of The Review:**

A good idea, well implemented and wrapped up in a quality paper

---

> ### Author Response · Authors · 2022-11-15
> **Response to Reviewer PWMt**
>
> Thank you for your review! We respond to your points here:
>
> ### Comparison to left-to-right generation in Table 11
>
> We theorize that the gap in performance between InCoder-6.7B and the higher-performing models in Table 11 is due to the larger-scale training of these models, rather than their use of a left-to-right objective. All models with higher HE@1 performance than InCoder-6.7B were trained on more Python code (in GB), used a larger model size, or both.
>
> In our Table 5, we perform a controlled comparison between causal masking training and standard left-to-right training with model and data size fixed, and find that causal masking training (CM) actually does slightly better than left-to-right training (LM) on HE@1 (rows 2 vs 3). This is also in line with the work of Bavarian et al. 2022, finding that a similar training procedure (FIM) did not substantially affect the model's left-to-right ability.
>
> ### Codex-2.5B results
>
> Thanks for your suggestion; we have added the Codex-2.5B results from Chen et al. to Table 11 in the updated version of the paper.

---

### Author Response · Authors · 2022-11-15
**General response to reviewers**

Thank you to all the reviewers for their time and feedback! We're glad that the reviewers found the paper to be a simple idea that nevertheless leads to new functionality (PWMt, 7zUC, CKVi), with experiments showing good performance (7zUC, bKjD, CKVi) on a thorough set of tasks (PWMt, 7zUC, CKVi), and that the paper was well-written (7zUC, bKjD, CKVi).

Several reviewers requested that the paper also discuss / compare to full code generation and a traditional left-to-right approach for training code models. While not the main focus of our paper, we have two comparisons and experiments that investigate this: (1) we compared our model to past left-to-right models on two full (left-to-right) Python function generation benchmarks (HumanEval and MBPP), finding that our model obtained similar performance to comparably-resourced left-to-right trained models [Table 11] and (2) we did find in our ablation section (Sec 5) that the traditional left-to-right training produced a model with similar (even slightly worse) left-to-right performance on HumanEval [Table 5]. We have added references to these sections in the Introduction to make this clearer.

We respond to the reviewers' other comments in threads on their reviews.

---

### Decision · Program_Chairs · 2023-01-20

**Decision:**

Accept: notable-top-25%

**Justification For Why Not Higher Score:**

The key training setting was proposed in an earlier work.

**Justification For Why Not Lower Score:**

The proposed idea was simple, straight-forward and led to good performance on a variety of tasks.  This is the kind of paper that we want to accept and see at ICLR.

**Metareview: Summary, Strengths And Weaknesses:**

This paper presents an approach to learn an in-filling model for code, by training with the causal masking setting proposed in (Aghajanyan et al., 2022a).  After training the model can be used to in-fill given left and right context significantly improving its performance on a variety of code modeling tasks, including in-filling code, generating documentation given code, type inference and variable name prediction.

The proposed idea is simple and straight-forward, and comes with a range of benefits.  The paper is well-written.  All reviewers liked the paper and favored acceptance.  The fact that the causal masking setting was proposed in prior work diminishes the novelty of this work somewhat, but it is still a good paper and I fully support its acceptance, given its appropriate fit with many code related tasks and good performance.

**Note From Pc:**

if the above contains the word "oral" or "spotlight" please see: "oral" presentation means -> notable-top-5% and "spotlight" means -> notable-top-25%. As stated in our emails, we are disassociating presentation type from AC recommendations